# Prenatal developmental toxicity of *Urtica simensis* essential oil in rat embryos and rat fetuses

Bickes Wube[1]*, Kaleab Asres[2], Samuel Woldekidan[3], Abiy Abebe[3], Yonas Girma[4], Girma Seyoum[1]

**1** Department of Anatomy, College of Health Sciences, Addis Ababa University, Addis Ababa, Ethiopia, **2** Department of Pharmaceutical Chemistry and Pharmacognosy, College of Health Sciences, Addis Ababa University, Addis Ababa, Ethiopia, **3** Traditional and Modern Medicine Research Directorate, Armauer Hansen Research Institute, Addis Ababa, Ethiopia, **4** Department of Pathology, College of Health Sciences, Addis Ababa University, Addis Ababa, Ethiopia

* bkswbe123@gmail.com

## Abstract

Pregnant women inhaled fresh *U. simensis* steam vapor to fumigate their bodies, and boiled it for tea. However, the safety of this practice during pregnancy has not yet been reported. This study aimed to evaluate the prenatal developmental toxicity of *U.simensis* essential oil in rat embryos and fetuses. The essential oil was extracted through hydrodistillation from aerial parts of *U.simensis*. One hundred pregnant rats were randomly assigned to five groups (20 rats per group). Groups I to III were administered oral doses of 250, 500, and 1000 mg/kg of *U.simensis* essential oil. Groups IV and V were used as pair-fed and ad libitum controls, respectively. The developing embryos and fetuses were retrieved on 12 and 20 days of gestation, respectively. Embryos were evaluated for growth and developmental delays. Fetuses were evaluated for external, skeletal, and visceral abnormalities. Oral doses of 250 and 500 mg/kg of *U.simensis* essential oil had no observed adverse effects in both rat embryos and rat fetuses. However, somite numbers and morphological scores were significantly decreased in gravid rats treated with 1000 mg/kg of essential oil. Embryonic developments of the caudal neural tube and forebrain were significantly delayed in pregnant dams administered 1000 mg/kg of essential oil. Crown-rump length and fetal weight were significantly decreased in gravid rats given 1000 mg/kg of essential oil. Gravid rats received 1000 mg/kg of essential oil also revealed a significant increase in fetal resorption. In conclusion, high-dose oral administration of *U.simensis* essential oil revealed detrimental effects in both rat embryos and fetuses. Therefore, pregnant women should be informed the potential risks associated with the nutraceutical use of *U.simensis* during pregnancy.

**Data availability statement:** The data supporting the findings were uploaded as Supporting information.

**Funding:** The author(s) received no specific funding for this work.

**Competing interests:** The authors have declared that no competing interests exist.

## Introduction

*Urtica simensis* Hochst. ex. A. Rich is a member of *Urtica* genus within *Urticaceae* family [1]. The *Urticaceae* family includes 54 genera with over 2000 species distributed in different regions such as tropical, subtropical and temperate climates worldwide [2–4]. This family contains the world's largest genus *Urtica* known as stinging nettle [5,6]. The *Urtica* genus encompasses more than 80 species distributed around the world [7,8]. *Urtica simensis* is endemic to Ethiopia and locally known as 'samma' in Amharic [9], 'Amei'e' in Tigrigna [10], Sanamik in Halaba [11], and 'Dobii' or 'Gurgubbee' in Oromifaa [12]. It is a dioecious and perennial nettle species [13]. In Ethiopia, the oval coarsely toothed leaves of the plant were consumed during drought and famine [14,15]. The dried *U.simensis* leaves were also used as flavors in a variety of foodstuffs [16,17]. In addition, this plant has been used for treatment of multiple ailments such as diabetes mellitus [18], gonorrhea [19], gastritis [12], heart failure [20], acute stomachache [21], peptic ulcer [22], wound infection [23], and malaria [24]. The essential oil isolated from aerial part of *U.simensis* showed anti-proliferative activity against ovarian and leukemia cancer cell lines [25]. In Ethiopia, pregnant women inhaled fresh *U.simensis* steam vapor to fumigate their body [9,26], powdered and boiled it for tea [27], roasted and ground it for juice [28], and ate it as stew [26]. When fresh herbs are boiled, essential oils present in the plants get released and inhaled [29]. Toxicological studies evaluate the safety of prenatal exposure to a substance and its effect on the developing organism [30,31]. The indiscriminate use of medicinal plants during pregnancy was exceedingly harmful [32]. A recent study about herbal medicine use during pregnancy reported that topical use of almond oil was substantially related to preterm birth, oral raspberry leaf with caesarean delivery, and excessive licorice consumption with early preterm birth [33]. Another recent study indicated that oral treatment of *U. simensis* leaf extract in pregnant rats revealed significant developmental delays and growth retardations [34]. Despite the reported nutraceutical potential of *U. simensis* essential oil, this study postulated that repeated oral intake at varying doses could adversely affect embryo-fetal development in rats. This hypothesis stems from its anti-proliferative effects on cancer cells, indicating a possible risk of prenatal toxicity. The Wistar albino rat was chosen as the animal model due to its widespread use as standard physiological and toxicological studies. The outcome of this study could contribute to the regulatory measures concerning utilization of this plant. Furthermore, the results would serve as groundwork for future investigations and devising strategies to manage its nutraceutical use among pregnant mothers.

## Materials and methods

### Plant collection

Fresh *U.simensis* were collected in August 2024 around Debremarkos town, located at GPS coordinates 10°20′ North, 37°43′ East, approximately 310 km Northwest of Addis Ababa, the capital city of Ethiopia. The aerial parts of well grown *U.simensis* were gathered at random using sanitized protective gloves. Identification and

authentication of the plant material were confirmed at the National Herbarium, Department of Plant Biology and Biodiversity Management, College of Natural and Computational Sciences, Addis Ababa University (AAU), where a voucher specimen (Collection number: bw-001) was deposited for future reference.

## Extraction

Fresh *U.simensis* aerial part (2 kg) were chopped into pieces and hydro-distilled for 3 hours using clevenger apparatus. The resultant condensate was isolated with 100 ml of hexane three times using a separatory funnel. To remove moisture, 15 g of anhydrous sodium sulfate was added to the hexane extract. Then, hexane was removed under vacuum at a temperature range of 30–35°C, resulting in a pale yellow essential oil that had an unpleasant odor. The extracted oil was kept in tightly sealed jar at −20 °C until it was used [35].

## GC-MS analysis

The GC-MS analysis was conducted using an Agilent Technologies 7890B gas chromatography (GC) and an Agilent Technologies 5977A Network mass spectrometry (MS). The GC used an Agilent Technologies HP-5MS non-polar column, measuring 30 m in length, 250 μm in internal diameter, and 0.25 μm in film thickness. A 1 μL sample was injected using a Gerstel cooled injection system, starting at −20 °C. The injector temperature was ramped to 320 °C during a 1:20 split injection. The GC separation procedure was carried out using hydrogen as carrier gas at a flow rate of 1.15 mL/min. Prior to injection, samples were diluted 1:1000 in methanol. The oven temperature program began at 40 °C (held for 3 minutes), then increased to 150 °C at 6 °C/min, followed by a ramp to 320 °C at 10 °C/min, and finally held at 320 °C for 3 minutes. Mass spectra were recorded using electron impact ionization at 70 eV in the range of 50 up to 550 atomic mass unit.

## Experimental animals

Healthy male and female (nulliparous, non-pregnant) Wistar albino rats, weighing 220–240 g and aged 10–12 weeks, were recruited from the animal breeding unit of Ethiopian Public Health Institute (EPHI). The animals had not previously undergone any experimental procedures. Rats of the same sex were housed in standard stainless steel cages, maintained under a 12-hour light and dark cycle, at room temperature of $23 \pm 3°C$ and relative humidity of $50 \pm 10\%$. The rats were provided a conventional laboratory diet and had free access to drinking water.

## Acute oral toxicity

Acute oral test was conducted on five female rats in accordance with OECD guideline 425 [36]. Initially, a single female rat was given 5000 mg/kg of essential oil through oral gavage. The remaining four female rats were administered 5000 mg/kg of essential oil 48 hours later since there were no fatalities in the first rat. As a control group, another five female rats were given the vehicle (distilled water with 2% Tween 80). After oral administration, clinical observation was conducted every 30 minutes for the first 4 hours and then daily for the next 14 days. Toxicity signs such as mortality, coma, convulsion, feces consistency, hair, itching, respiration, salivation and other behavioral patterns were assessed.

## Mating, grouping and administration

After one week of acclimatization, rats were mated overnight by placing male rats in a cage with virgin female rats in 1:1 ratios. Female rats were checked next morning for copulatory plug and vaginal smears were taken for microscopic examination of spermatozoa. The first day detection of spermatozoa in the vaginal swab was taken as day zero of pregnancy [37]. Thereafter, pregnant rats were randomly assigned to five groups, with each group consisting of 20 gravid rats, in accordance with OECD guidelines [38,39]. Groups I-III were administered 250, 500 and 1000 mg/kg of essential oil, respectively, based on the acute oral test. Groups IV and V were pair-fed and ad libitum control, respectively. The

administration was given through oral gavage from 6th to 12th days of gestation since it was a critical period of organogenesis in rats [40]. The pair-fed control rats received only the vehicle (distilled water with 2% Tween 80) at 1 ml/100 g body weight. The ad libitum control rats were fed freely and kept untouched during the experiment.

### Clinical observation, dietary intake and body weight

The clinical observation of pregnant rats was made daily at the same time for all signs of overt toxicity including mortality, morbidity and pertinent behavioral changes. Their dietary intake was weighed every morning until the end of the experiment. Each pregnant dam's weight was taken at 1st confirmation of pregnancy, 6th, 12th and 20th days of gestation [41].

### Embryo evaluation

On the 12th day of gestation, at noon, fifty pregnant rats (ten per group) were anesthetized using an intraperitoneal injection of sodium pentobarbital (150 mg/kg) [42]. Subsequently, they were placed supine on an operating table. The extremities were stretched out and set in place, and a vertical abdominal skin incision was made to reveal the abdominal cavity. Pins were applied to hold skin flaps and abdominal muscles on both sides. The uterine horns were removed and placed in Hank's solution. In order to expose the developing embryos, uterine horns were incised along the anti-mesometrial boundary [43,29]. The membranes of the embryos were dissected with fine forceps to show the underlying visceral yolk sac. The development and circulation of yolk were looked through dissecting microscope. The progression of each developmental system was assessed using the Brown and Fabro criteria, which contained 16 noticed embryonic developmental endpoints [44,45]. There were up to six developmental stages, and a score ranging from 0 to 5 was assigned for each developmental endpoint. The crown-rump length (CRL) of each embryo was measured, and the morphological score was also computed.

### Fetal evaluation

Similar to the evaluation of embryos, fifty near-term pregnant rats (ten per group) on the 20th day of gestation were dissected using a vertical abdominal incision [42]. The uterine horns were fully exposed and examined. The metrial glands, which are yellowish nodules along the mesometrial border of uterine horns, were counted to assess implantation sites. The number of prior resorptions was reflected by metrial nodules that were not occupied by living or recently deceased fetuses. Gently pressing on the fetus revealed whether it was alive or dead. The uterine horns have been cut along the anti-mesometrial border to reveal fetuses, fetal membranes, and placentas. The fetuses were explanted, and the placenta was removed. The CRL of each fetus was measured from the top of the head to the buttocks. Each fetus's anogenital distance (AGD) was measured from the cranial edge of the anus to the caudal border of the genital tubercle. Each fetus's placental weight was also measured.

### External and visceral examination

Each fetus underwent examination for craniofacial developmental anomalies, limb developmental abnormalities, vertebral column malformations, tail development disorders and abnormalities of the external genitalia. After external examination, some fetuses from each pregnant rat were preserved for skeletal staining. The remaining fetuses were fixed in Bouin's solution for visceral examination [46]. These Bouin's solution-fixed fetuses were serially sectioned using a dissecting microscope and a modified Wilson technique [47]. The craniocaudal section was done through the jaw and posteriorly above the ear and the presence of palatine cleft was assessed. Coronal section of the head and transverse section of the rest of the body were also done to assess any readily apparent abnormalities in the brain, neck, thorax and abdominopelvic regions.

## Skeletal staining

Two to three fetuses from each gravid rat were placed in 95% ethanol for one week. These alcohol-preserved fetuses were eviscerated through midline incision and underwent skeletal staining using revised techniques of Rigueur and Lyons [48]. Following alcohol preservation, the fetuses underwent a clearing process using 1% potassium hydroxide (KOH) solution until the bones became transparent, which typically took two days. Thereafter, the fetuses were transferred to fresh 1% KOH solution and stained with a few drops (0.4 ml) of alizarin red. The staining process was allowed to proceed overnight, and any excess staining was addressed by placing the specimens in a Mall's solution (79% distilled water, 20% glycerin, and 1% KOH). The specimens were placed in a gradual transition through increasing concentrations of glycerin (20%, 40%, 60%, and 80%) over the course of approximately one week for each concentration and stored in 100% glycerin. A small thymol crystal was added as a precautionary measure to prevent fungal growth and contamination. Finally, the skeletal scoring chart developed by Nash and Persaud was used to evaluate skeletal ossification [49]. The sternum, metacarpal, metatarsal and sacrococcygeal bones degree of ossification were taken as the key markers of skeletal development in rats [50]. The ossifications of the skull, hyoid, ribs, vertebrae and limb bones were also examined.

## Gross examination of placenta and histopathology

The placenta of each fetus was examined for any visible abnormalities in its structure and appearance. Placental tissues were taken from 2 to 3 fetuses per dam for histopathological examination. Each tissue sample was taken in 3–4 mm sections and fixed with 10% formalin. After fixation, tissue samples underwent a dehydration process using an ascending series of alcohol concentrations (40%, 50%, 70%, 80%, 90% and 100%). The dehydrated tissues were cleared using xylene solutions. Following clearing, tissue samples were infiltrated with melted paraffin wax. Finally, each sample was placed into an embedding cassette and surrounded with more melted paraffin wax to create a solid wax block containing the tissue. Then thin sections, measuring 5 μm in thickness, were cut from each paraffin block using a microtome. These tissue sections were carefully placed onto frosted glass slides and kept in a warm oven at 40–45°C for 20–30 minutes [51].

For staining, tissue sections were mounted on the slides first and underwent a dewaxing process. The slides were immersed in three separate xylene solutions for 5 minutes each to remove the embedding paraffin wax. Next, the tissues were rehydrated by passing them through a descending series of alcohol concentrations. The slides were submerged in absolute alcohol (two steps), then 90%, 80% and 70% alcohol solutions for 2 minutes each. After the rehydration, the slides were washed under running tap water. They were then stained with Harris hematoxylin for 5–10 minutes. The slides were rinsed in tap water for 10 minutes to remove excess hematoxylin. They were then briefly dipped in acid alcohol for 2–3 seconds for differentiation. Following the hematoxylin staining, the slides were counterstained with eosin Y for 1–2 minutes. To complete the staining procedure, the slides underwent dehydration again using an ascending series of alcohol concentrations (80%, 95%, and two steps of absolute alcohol). The slides were then cleared using xylene. Finally, the stained and dehydrated tissue sections were mounted with synthetic resin called DPX (Dibutylphthalate Polystyrene Xylene) and sealed with coverslips for microscopic examination [51]. The stained placental tissue slides were examined by an experienced pathologist using a binocular light microscope. The pathologist assesses the structural integrity and features of the different zones and cell types within the placenta. Observations through the microscope were photographed using the microscope's automated built-in digital camera (a Leica EC4 model, Germany). The photographs were captured at 10x objective lens magnifications.

## Statistical analysis

Data were cleaned, coded and entered using Epi-data version 3.1. Then, it was exported to statistical package for social science (SPSS) version 25 for analysis. The Shapiro-Wilk and Levene's tests were used to assess normality of data and homogeneity of variance, respectively. Then, one-way analysis of variance (ANOVA) with Tukey's post hoc test was done

for comparison within and between treatment and control groups. In cases where the data did not meet the assumptions of ANOVA, the Kruskal-Wallis test was applied for the analysis. The data were reported as mean ± standard deviation. The findings were considered statistically significant if the p-value was less than 0.05.

### Ethical statement

Ethical approval was obtained from Addis Ababa University, College of Health Sciences, with protocol number 045/22/anatomy in compliance with OECD guidelines for the care and use of animals [52]. The housing, care, and use of experimental animals were also conducted in accordance with the European Union guidelines outlined in Directive 2010/63/EU [53]. All experimental procedures were carried out in line with the ARRIVE guidelines (Animal Research Reporting of In Vivo Experiments) [54]. All animal dissection was performed under sodium pentobarbital anesthesia based on the American Veterinary Medical Association (AVMA) guidelines [55], and all efforts were made to minimize animal suffering. At the end of the experiment, carcasses of sacrificed rats and any unused fetal specimens were disposed of in a humane manner following the laboratory standards and protocols of EPHI.

## Results

### Chemical compositions of *U.simensis* essential oil

GC-MS analysis of *U.simensis* essential oil revealed 43 compounds, accounting for 67.69% of the total composition. The most abundant constituents were 3, 5-dimethyl-1, 2, 4-trithiolane (12.5%), followed by cis-3-hexen-1-ol (5.62%), linalool (5.41%), citronellol (5.10%), methyl chavicol (4.50%), and cis-3-hexenyl acetate (4.20%) (Table 1).

### Acute effect of *U.simensis* essential oil

No mortality was observed in any of the animals treated with a single oral dose of 5000 mg/kg of essential oil. Daily clinical observations did not show any signs of toxicity. In addition, there were no significant differences in daily dietary intake or weight gain between essential oil-treated rats and the vehicle control rats. The median lethal dose ($LD_{50}$) of essential oil exceeded 5000 mg/kg body weight.

### Clinical observation of pregnant dams

Daily cage-side clinical observations indicated that there were no instances of abortion or maternal death in any of the experimental groups. All pregnant dams showed no signs of overt toxicity or morbidity.

### Effects of *U.simensis* essential oil on dietary intake and body weight

Pregnant dams received essential oil demonstrated a dose-dependent decrease in daily dietary intake compared to the control groups, but it was not statistically significant. However, pregnant dams treated with 1000 mg/kg of essential oil experienced significantly lower weight gain compared to the pair-fed and ad libitum control groups (Table 2).

### Effects of *U.simensis* essential oil on embryonic growth and development

No statistically significant differences were observed in the crown-rump length (CRL) of embryos between treatment and control groups. However, litters from dams treated with 1000 mg/kg of the essential oil exhibited a significantly lower mean somite number and morphological score compared to both pair-fed and ad libitum controls (Table 3).

In vivo embryonic development was evaluated by assessing morphological endpoints in the circulatory, neurological, musculoskeletal and craniofacial systems (Fig 1). The embryo developmental scores exhibited a dose-dependent decrease for each system (Tables 4 and 5). Notably, gravid dams administered 1000 mg/kg of the essential oil showed significantly reduced developmental scores in the caudal neural tube and forebrain compared to both pair-fed and ad libitum control groups.

**Table 1. Chemical compositions of *U.simensis* essential oil identified through GC-MS analysis.**

| Formula | Name | Area (%) | RI | Formula | Name | Area (%) | RI |
|---|---|---|---|---|---|---|---|
| $C_8H_{16}O$ | 1-Octen-3-oL | 0.23 | 970 | $C_8H_{14}O$ | 6-Methyl-5-hepten-2-one | 3.25 | 984 |
| $C_{10}H_{22}$ | n-Decane | 0.29 | 989 | $C_8H_{14}O_2$ | cis-3-Hexenyl acetate | 4.20 | 1008 |
| $C_9H_{12}O_2$ | 4-Ketoisophorone | 0.33 | 1137 | $C_{10}H_{16}O$ | Camphor | 0.16 | 1138 |
| $C_{11}H_{10}$ | 1-Methyl-naphthalene | 1.21 | 1300 | $C_{11}H_{10}$ | 2-Methyl-naphthalene | 0.50 | 1317 |
| $C_9H_{20}O$ | n-Nonanol | 0.68 | 1159 | $C_{10}H_{18}O$ | α-Terpineol | 0.53 | 1180 |
| $C_{10}H_{12}O$ | Methyl chavicol | 4.50 | 1209 | $C_{10}H_{16}O$ | trans-4-caranone | 0.44 | 1175 |
| $C_7H_{12}O$ | cis-4-Heptenal | 0.75 | 897 | $C_7H_{14}O$ | Heptanal | 0.20 | 905 |
| $C_{12}H_{24}O_2$ | Dodecanoic acid | 0.26 | 1554 | $C_{15}H_{24}O$ | Caryophyllene oxide | 4.32 | 1576 |
| $C_{10}H_{18}O$ | 1,8-Cineole | 0.28 | 1020 | $C_7H_{14}O$ | cis-4-Hepten-1-ol | 0.26 | 960 |
| $C_{15}H_{26}O$ | Bulnesol | 0.67 | 1675 | $C_{15}H_{26}O$ | 4-*epi-cis*-Dihydroagarofuran | 0.15 | 1504 |
| $C_{13}H_{20}O$ | β-Ionone | 1.22 | 1493 | $C_6H_{14}O$ | 1-Hexanol | 0.18 | 874 |
| $C_6H_{12}O$ | cis-3-Hexen-1-ol | 5.62 | 860 | $C_{10}H_{18}O$ | Linalool | 5.41 | 1090 |
| $C_{10}H_{16}O$ | Fenchone | 1.56 | 1079 | $C_4H_8S_3$ | 3,5-Dimethyl-1,2,4-trithiolane | 12.35 | 1140 |
| $C_{10}H_{14}$ | 1,3,8-p-Menthatriene | 0.52 | 1120 | $C_{10}H_{18}O$ | Geraniol | 1.33 | 1259 |
| $C_{10}H_{20}O$ | Citronellol | 5.10 | 1219 | $C_6H_{10}O$ | trans-2-hexenal | 0.20 | 850 |
| $C_4H_8S_4$ | 3,6-dimethyl-1,2,4,5-tetrathiane | 0.25 | 1398 | $C_8H_{18}O$ | n-Octanol | 0.24 | 1066 |
| $C_9H_{10}O_2$ | p-Vinyl guaiacol | 3.28 | 1312 | $C_6H_{12}S_3$ | 2,4,6-Trimethyl-1,3,5-trithiane | 1.35 | 1302 |
| $C_{10}H_{14}O$ | Pinocarvone | 0.23 | 1667 | $C_{10}H_{18}O$ | Borneol | 0.15 | 1161 |
| $C_{10}H_{12}O$ | p-Ethylacetophenone | 0.62 | 1275 | $C_{12}H_{18}O_2$ | Thymohydroquinone dimethyl ether | 0.38 | 1449 |
| $C_{10}H_{12}O_2$ | Eugenol | 2.82 | 1360 | $C_{11}H_{18}O_2$ | Neryl formate | 0.24 | 1285 |
| $C_7H_{16}O$ | n-Heptanol | 0.24 | 950 | | | | |
| $C_8H_8$ | Styrene | 0.71 | 879 | | | | |
| $C_{12}H_{12}$ | 1,3-Dimethyl-naphthalene | 0.48 | 1422 | | | | |

Total identified = 67.69%, Total unidentified = 32.31%

RI – retention indices.

**Table 2. Dietary intake and weight gain of pregnant dams treated with *U. simensis* essential oil.**

| Groups | Food intake (g/day) | | Water intake (ml/day) | | Weight gain/dam(g) | |
|---|---|---|---|---|---|---|
| | Day-12 | Day-20 | Day-12 | Day-20 | Day-12 | Day-20 |
| Group I – 250 mg/kg | 199.74 ± 0.88 | 200.97 ± 0.48 | 26.67 ± 1.16 | 27.33 ± 1.56 | 14.63 ± 0.64 | 18.20 ± 2.64 |
| Group II – 500 mg/kg | 199.48 ± 1.05 | 200.72 ± 0.48 | 26.42 ± 1.31 | 27.08 ± 1.44 | 13.36 ± 0.51 | 17.90 ± 2.53 |
| Group III – 1000 mg/kg | 199.23 ± 0.95 | 200.71 ± 0.90 | 26.42 ± 1.50 | 26.92 ± 1.44 | 10.60 ± 0.55* | 14.50 ± 2.42* |
| Group IV – vehicle | 200.20 ± 1.25 | 200.97 ± 0.93 | 26.75 ± 1.05 | 27.42 ± 1.44 | 14.15 ± 2.25 | 18.40 ± 2.32 |
| Group V- ad libitum | 199.91 ± 1.01 | 201.10 ± 0.82 | 26.92 ± 1.17 | 27.75 ± 1.37 | 14.20 ± 1.05 | 18.60 ± 2.00 |

Data were written as mean ± SD (standard deviation),*significant differences as compared to pair fed and ad libitum controls, one way ANOVA.

## Effects of *U.simensis* essential oil on fetal outcomes

No dead fetuses were found in any of the experimental groups. The mean number of implantation and resorption sites per litter was computed and analyzed for both treatment and control groups. Pregnant dams treated with 1000 mg/kg of essential oil showed a statistically significant increase in resorption per litter compared to both the pair-fed and ad libitum control groups. However, there was no statistically significant difference in implantation sites per litter between the gravid rats treated with essential oil and the control groups (Table 6 and Fig 2).

**Table 3. Embryonic growth indices of rat embryos treated with *U.simensis* essential oil.**

| Groups | Number of embryos | CRL/litter (cm) | Somite number/litter | Morphological score/litter |
|---|---|---|---|---|
| Group I – 250 mg/kg | 91 | 3.80 ± 0.01 | 29.13 ± 1.33 | 40.79 ± 1.08 |
| Group II – 500 mg/kg | 89 | 3.79 ± 0.02 | 29.12 ± 1.31 | 40.76 ± 1.12 |
| Group III – 1000 mg/kg | 90 | 3.79 ± 0.02 | 27.91 ± 1.04* | 39.60 ± 1.03* |
| Group IV – pair fed | 90 | 3.80 ± 0.02 | 29.19 ± 1.36 | 40.87 ± 1.04 |
| Group V- ad libitum | 91 | 3.80 ± 0.02 | 29.18 ± 1.31 | 40.90 ± 1.07 |

Data were written as mean ± SD (standard deviation), CRL = crown rump length, *significant difference as compared to pair fed and ad-libitum controls, one way ANOVA.

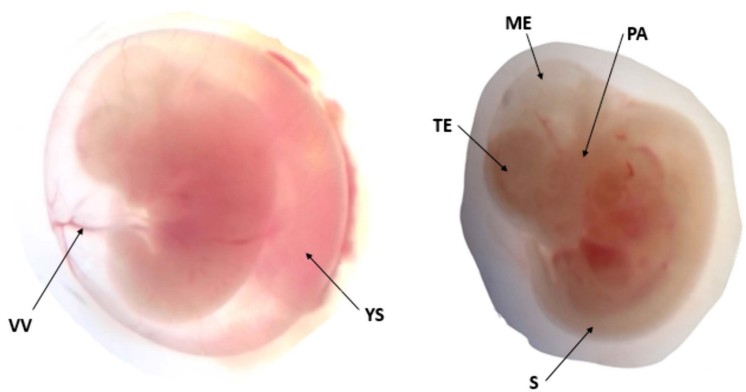

**Fig 1. In vivo development of 12-day old rat embryos treated with *U.simensis* essential oil: VV – vitelline blood vessels, YS-yolk sac, TE- telencephalon, ME-mesencephalon, PA-pharyngeal arch, S-somites.**

**Table 4. Developmental scores of the nervous system of rat embryos treated with *U.simensis* essential oil.**

| Groups | Caudal neural tube | Hind brain | Mid brain | Fore brain | Otic system | Optic system | Olfactory system |
|---|---|---|---|---|---|---|---|
| Group I – 250 mg/kg | 3.08 ± 0.67 | 2.82 ± 0.70 | 2.56 ± 0.77 | 3.01 ± 0.68 | 2.99 ± 0.64 | 2.49 ± 0.67 | 2.88 ± 0.71 |
| Group II – 500 mg/kg | 3.03 ± 0.69 | 2.76 ± 0.69 | 2.48 ± 0.74 | 2.90 ± 0.67 | 2.93 ± 0.60 | 2.44 ± 0.62 | 2.82 ± 0.68 |
| Group III – 1000 mg/kg | 2.83 ± 0.71* | 2.72 ± 0.70 | 2.44 ± 0.72 | 2.81 ± 0.69* | 2.84 ± 0.65 | 2.41 ± 0.63 | 2.77 ± 0.70 |
| Group IV – pair fed | 3.14 ± 0.68 | 2.97 ± 0.73 | 2.53 ± 0.72 | 3.11 ± 0.68 | 3.07 ± 0.60 | 2.59 ± 0.70 | 2.96 ± 0.73 |
| Group V – ad libitum | 3.12 ± 0.70 | 2.96 ± 0.74 | 2.68 ± 0.83 | 3.14 ± 0.67 | 3.09 ± 0.63 | 2.64 ± 0.74 | 3.00 ± 0.76 |

Data were written as mean ± SD (standard deviation), *significant difference as compared to pair fed and ad-libitum controls, one way ANOVA.

**Table 5. Developmental scores of cardiovascular and musculoskeletal systems of rat embryos treated with *U.simensis* essential oil.**

| Groups | Yolk sac circulation | Heart | Flexion | Branchial bars | Maxillary process | Mandibular process | Fore limb | Hind limb |
|---|---|---|---|---|---|---|---|---|
| Group I – 250 mg/kg | 3.05 ± 0.71 | 2.84 ± 0.64 | 2.40 ± 0.50 | 2.98 ± 0.61 | 1.58 ± 0.49 | 0.84 ± 0.37 | 2.16 ± 0.76 | 1.59 ± 0.49 |
| Group II – 500 mg/kg | 3.01 ± 0.75 | 2.78 ± 0.65 | 2.35 ± 0.48 | 2.90 ± 0.64 | 1.51 ± 0.50 | 0.78 ± 0.42 | 2.11 ± 0.74 | 1.53 ± 0.50 |
| Group III – 1000 mg/kg | 2.99 ± 0.71 | 2.76 ± 0.64 | 2.33 ± 0.52 | 2.88 ± 0.63 | 1.49 ± 0.50 | 0.73 ± 0.44 | 2.10 ± 0.73 | 1.52 ± 0.50 |
| Group IV – pair fed | 3.10 ± 0.74 | 2.88 ± 0.67 | 2.44 ± 0.50 | 3.04 ± 0.65 | 1.59 ± 0.49 | 0.84 ± 0.36 | 2.24 ± 0.74 | 1.64 ± 0.51 |
| Group V – ad libitum | 3.14 ± 0.80 | 2.90 ± 0.66 | 2.46 ± 0.50 | 3.07 ± 0.64 | 1.62 ± 0.48 | 0.85 ± 0.36 | 2.27 ± 0.73 | 1.68 ± 0.55 |

Data were written as mean ± SD (standard deviation).

**Table 6. Fetal outcomes of gravid rats treated with *U.simensis* essential oil.**

| Groups | Live fetuses | Dead fetuses | Implantation sites/litter | Resorption sites/litter | Male fetuses/dam | Female fetuses/dam |
|---|---|---|---|---|---|---|
| Group I – 250 mg/kg | 92 | 0 | 9.20 ± 0.78 | 0.40 ± 0.52 | 4.30 ± 1.45 | 4.80 ± 1.23 |
| Group II – 500 mg/kg | 90 | 0 | 9.00 ± 0.83 | 0.50 ± 0.71 | 4.20 ± 0.79 | 4.80 ± 0.78 |
| Group III – 1000 mg/kg | 91 | 0 | 9.10 ± 0.87 | 1.30 ± 1.25* | 4.10 ± 1.37 | 5.00 ± 1.05 |
| Group IV – Pair fed | 92 | 0 | 9.20 ± 0.1.14 | 0.30 ± 0.48 | 4.40 ± 1.71 | 4.80 ± 1.68 |
| Group V – ad libitum | 93 | 0 | 9.30 ± 1.34 | 0.20 ± 0.42 | 4.40 ± 1.58 | 4.90 ± 1.91 |

Data were written as mean ± SD (standard deviation), *significant difference as compared to pair fed and ad-libitum controls, one way ANOVA.

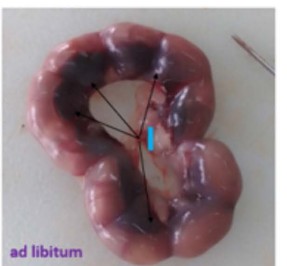
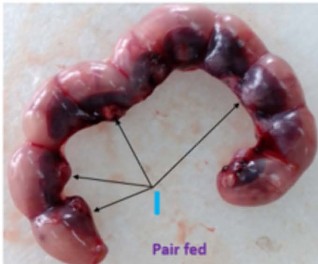
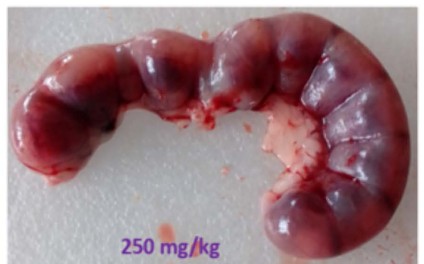
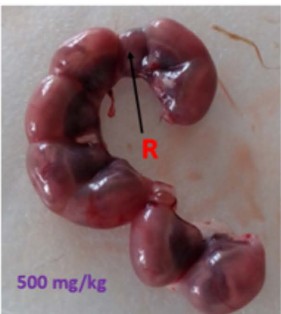
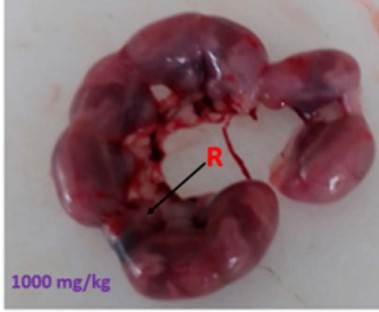
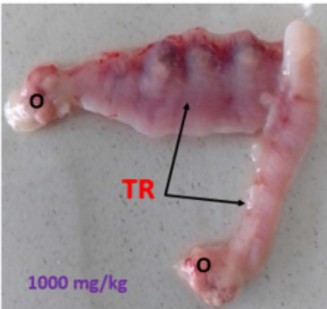

**Fig 2. Implantation and resorption of 20-day old rat fetuses treated with *U.simensis* essential oil I-implantation, R-resorption, TR-total resorption only in one gravid rat, O-ovary.**

### Effects of *U.simensis* essential oil on fetal growth and androgen dependent endpoints

The mean fetal weight in gravid rats treated with 1000 mg/kg of essential oil was 3.47 ± 0.18 g, significantly lower than that of the pair-fed (3.55 ± 0.17 g) and ad libitum (3.56 ± 0.18 g) control groups. The mean CRL was also significantly lower in the high-dose treated group (4.54 ± 0.18 cm) compared to the pair-fed (4.62 ± 0.15 cm) and ad libitum (4.63 ± 0.17 cm) controls. However, there was no statistically significant difference in the mean weight of fetal placentas between the treatment and control groups. The anogenital distance, presence of multiple nipples/areola and external genital abnormalities were used to assess the essential oil's androgen related effects. However, there were no visible abnormalities in the nipples, areola, and external genitalia. There were also no significant differences in AGD between treatment groups exposed to essential oil and control groups in both male and female fetuses (Table 7).

## Effects of *U.simensis* essential oil on fetal external and visceral structural formations

The explanted fetuses were assessed from head to tail for treatment related developmental defects (Fig 3). Limb deformities, tail abnormalities, craniofacial malformations, vertebral column discrepancies, and obvious genital defects were examined, but no discrete external anomaly was observed.

In addition to external examination, the fetuses fixed in Bouin's solution were serially sectioned and examined for visceral anomalies (Fig 4). The head, neck, chest, abdomen and abdominopelvic regions were thoroughly inspected for any anomalies using a dissecting microscope. Conditions such as cleft palate, hydrocephalus, eye-related anomalies, and abnormalities in the neck, trachea, and cardiac septum were assessed in the head and neck regions. Additionally, diaphragmatic hernia, agenesis of abdominal viscera, and external genitalia were also examined. However, no visible visceral abnormalities were observed through the dissecting microscope.

## Effects of *U.simensis* essential oil on fetal skeletal ossifications

As shown in Table 8, the mean ossification centers in the axial and extremity bones of 20-day-old rat fetuses treated with essential oil were counted and analyzed. However, there was no significant difference in skeletal ossification between the essential oil-treated and control animals, whether pair-fed or ad libitum (Fig 5).

## Effects of *U.simensis* essential oil on histology of the placenta

A microscopic analysis of the placenta was performed to evaluate any histological irregularities, such as necrosis and hemorrhage (Table 9). However, no histological abnormalities were observed in any of the experimental groups (Fig 6).

**Table 7. Fetal growth, anogenital distance and placental weight of rat fetuses treated with *U.simensis* essential oil.**

| Groups | Fetal weight (g) | CRL (cm) | Female AGD (mm) | Male AGD (mm) | Placental weight (g) |
|---|---|---|---|---|---|
| Group I – 250 mg/kg | 3.53±0.17 | 4.58±0.16 | 4.44±0.78 | 6.02±0.62 | 0.54±0.06 |
| Group II – 500 mg/kg | 3.52±0.16 | 4.56±0.16 | 4.42±0.71 | 5.95±0.66 | 0.53±0.06 |
| Group III – 1000 mg/kg | 3.47±0.18* | 4.54±0.18* | 4.40±0.75 | 5.73±0.63 | 0.52±0.06 |
| Group IV – pair fed | 3.55±0.17 | 4.62±0.15 | 4.48±0.82 | 5.93±0.72 | 0.54±0.06 |
| Group V – ad libitum | 3.56±0.18 | 4.63±0.17 | 4.49±0.80 | 6.05±0.60 | 0.55±0.07 |

Data were written as mean±SD (standard deviation), *significant difference compared to pair fed and ad libitum controls, one way ANOVA, CRL=crown rump length, AGD=anogenital distance.

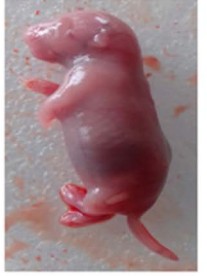 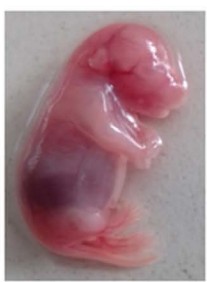 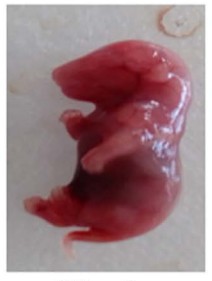 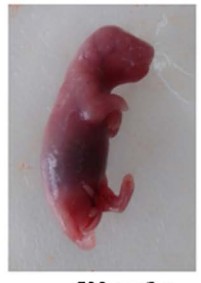 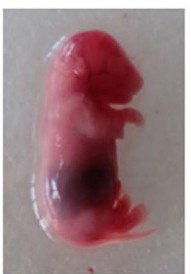

ad libitum          Pair fed          250 mg/kg          500 mg/kg          1000 mg/kg

**Fig 3. External examination of 20-day old rat fetuses treated with *U.simensis* essential oil.**

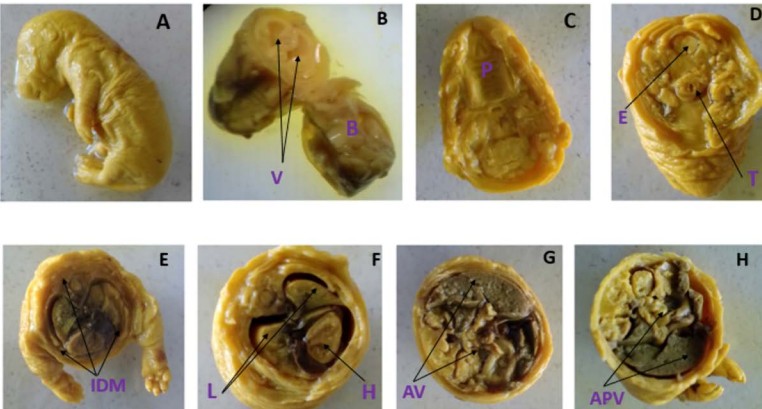

**Fig 4. Visceral examination of Bouin's solution fixed 20-day old rat fetuses treated with *U.simensis* essential oil:** (A) un-sectioned fetus, (B) coronal section of head, V- ventricles, B- brain tissue, (C) section made through jaw and above ear, P-palate, (D) transverse section of neck, E-esophagus, T-trachea, (E) transverse section of thorax, IDM-intact diaphragm, (F) mediastinal viscera, L-lung, H-heart, (G) horizontal section of abdomen, AV-abdominal viscera-liver and intestine, (H) horizontal section of abdominopelvic region, APV- abdominopelvic viscera.

**Table 8. Axial and extremity bone ossification centers of rat fetuses treated with *U. simensis* essential oil.**

| Groups | Number of ossification centers per group | | | | | | | | |
|---|---|---|---|---|---|---|---|---|---|
| | Sternum | Ribs | Thoracic | Lumbar | Caudal | FL phalanges | HL phalanges | Metacarpals | Metatarsals |
| Group I – 250 mg/kg | 4.00 ± 0.71 | 24 ± 0.0 | 12 ± 0.0 | 5 ± 0.0 | 3.92 ± 0.57 | 3.72 ± 0.45 | 3.64 ± 0.49 | 4.12 ± 0.44 | 3.84 ± 0.47 |
| Group II – 500 mg/kg | 3.96 ± 0.74 | 24 ± 0.0 | 12 ± 0.0 | 5 ± 0.0 | 3.88 ± 0.53 | 3.68 ± 0.47 | 3.56 ± 0.50 | 4.08 ± 0.40 | 3.80 ± 0.41 |
| Group III – 1000 mg/kg | 3.84 ± 0.62 | 24 ± 0.0 | 12 ± 0.0 | 5 ± 0.0 | 3.80 ± 0.64 | 3.60 ± 0.50 | 3.52 ± 0.51 | 4.04 ± 0.54 | 3.76 ± 0.44 |
| Group IV – pair fed | 4.08 ± 0.64 | 24 ± 0.0 | 12 ± 0.0 | 5 ± 0.0 | 4.04 ± 0.54 | 3.72 ± 0.46 | 3.60 ± 0.50 | 4.16 ± 0.55 | 3.84 ± 0.37 |
| Group V – ad libitum | 4.20 ± 0.65 | 24 ± 0.0 | 12 ± 0.0 | 5 ± 0.0 | 4.12 ± 0.52 | 3.76 ± 0.43 | 3.68 ± 0.48 | 4.20 ± 0.64 | 3.88 ± 0.33 |

Results are written as mean ± SD (standard deviation), FL = Fore limb, HL = Hind limb.

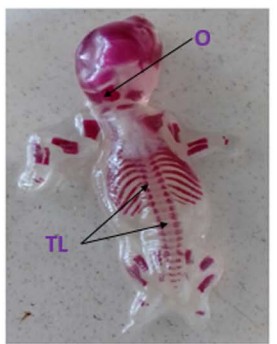
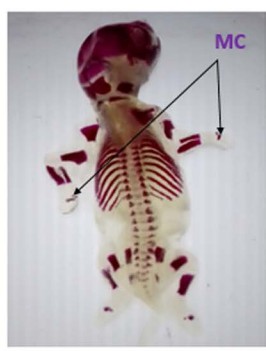
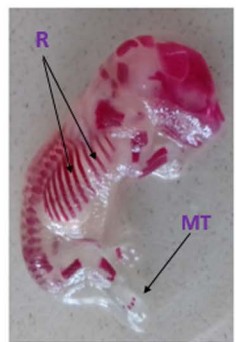
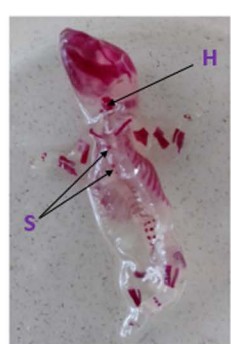

**Fig 5. Skeletal ossification of alizarin red stained 20-day old rat fetuses treated with *U.simensis* essential oil:** TL-thoracic and lumbar vertebrae, O-occipital bone, MC-metacarpals, R- ribs, MT-metatarsals, H-hyoid, S-sternum.

**Table 9. Histopathological scoring of fetal placenta exposed to *U. simensis* essential oil.**

| Groups | Necrosis | Hemorrhage | Intervillous thrombosis | Calcification | Vascular dilatation |
|---|---|---|---|---|---|
| Group I – 250 mg/kg | 0 | 0 | 0 | 0 | 0 |
| Group II – 500 mg/kg | 0 | 0 | 0 | 0 | 0 |
| Group III – 1000 mg/kg | 0 | 0 | 0 | 0 | 0 |
| Group IV – Pair fed | 0 | 0 | 0 | 0 | 0 |
| Group V – ad libitum | 0 | 0 | 0 | 0 | 0 |

Zero (0) values indicate no pathological lesion for each fetal placenta.

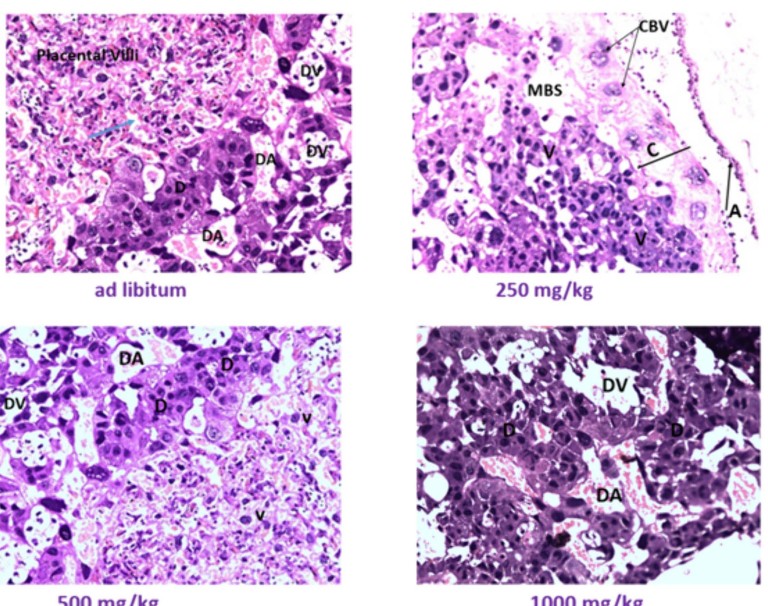

**Fig 6. Microscopic examination of fetal placenta exposed to *U.simensis* essential oil: D-decidua, DA-decidual artery, DV-decidual vein, blue arrow head-maternal blood space, CBV-chorionic blood vessels, MBS-maternal blood vessels, V-each placental villus, C-chorion, A-amnion.**

## Discussion

The pharmaceutical sector is increasingly looking for natural products with biological properties [56]. Essential oils are volatile and fragrant extracts derived from different parts of aromatic plants such as flowers, leaves, stems, bark or roots. These oils are renowned for their distinct fragrances and have been used for centuries in various cultures for their therapeutic, cosmetic and culinary purposes [57–59]. Major components of *U. simensis* essential oil were aromatic hydrocarbons, organosulfur and monoterpenoid compounds [25]. Aromatic hydrocarbons were principal groups of substances found in essential oils and responsible for its distinct aroma. Organosulfur compounds and monoterpenoid were also important ingredients adding to the overall makeup and possible medicinal qualities of essential oils [59,60]. Although *U.simensis* essential oil was used as a therapeutic agent and for nutritional value [25], its safety in prenatal exposure has not been reported yet. This study aimed to evaluate the prenatal developmental toxicity of *U.simensis* essential oil in rat embryos and rat fetuses. Pregnant dams were administered *U.simensis* essential oil through gavage at doses of 250, 500 and 1000 mg/kg from the 6th to 12th days of gestation. The findings of this study showed that oral administration of 250 and 500 mg/kg of *U.simensis* essential oil did not show any observable adverse effects in both rat embryos

and fetuses. However, gravid rats treated with 1000 mg/kg of *U.simensis* essential oil revealed significant developmental delays and growth retardations in both rat embryos and fetuses. The somite and morphological scores in 12-day-old rat embryos exposed to 1000 mg/kg of *U.simensis* essential oil were significantly low as compared to those of pair fed and ad libitum control groups. The mean crown-rump length of developing embryos treated with *U.simensis* essential oil was low as compared to the control groups, but it was not statistically significant. The significant decrease in somite and morphological scores might be due to the presence of naphthalene, eugenol and organosulfur compounds found in *U. simensis* essential oil, as reported by Ayalew et al. [25], which deter cell cycle [61–63]. The dose dependent reduction in the embryonic growth indices might also be due to the presence of alkaloids in *U.simensis* essential oil, especially linalool, an acyclic monoterpenoid known to inhibit growth and development [64–66]. In this study, the in vivo developments of the caudal neural tube and forebrain in 12-day old rat embryos exposed to 1000 mg/kg of *U.simensis* essential oil was significantly delayed compared to the pair fed and ad libitum control groups. These could be attributed to the presence of tannins and other alkaloids in the *U.simensis* essential oil which halted growth and development [67,68]. External and visceral morphological assessments of 20 days old rat fetuses revealed no noticeable treatment related anomalies in the cranial, nasal, oral, trunk and visceral organs. However, fetuses exposed to 1000 mg/kg of *U.simensis* essential oil exhibited significant reduction in crown rump length and fetal weight compared to both pair fed and ad libitum control groups. In addition, the mean fetal resorption significantly increased in gravid rats treated with 1000 mg/kg of *U. simensis* essential oil compared to the control groups. The reduction of crown rump length and fetal weight might be due to the presence of some constituents of *U.simensis* essential oil, especially 1,8-cineole and eugenol, which deter cell proliferation, as evidenced by anti-proliferative properties of cancer cell lines [25,69]. Some studies also reported that exposure to aromatic hydrocarbons showed developmental toxicity [70,71]. The increased fetal resorption in the 1000 mg/kg treated dams could be attributed to the apoptosis role of the chemical compositions of *U. simensis* essential oil [64] and the role of polycyclic aromatic hydrocarbon constituents of *U. simensis* essential oil as an intermediary effect of infertility [71]. Nevertheless, the anogenital distance in 20-day-old male and female fetuses showed no significant difference between gravid rats exposed to *U. simensis* essential oil and control groups. Furthermore, no abnormalities were observed in the nipples, areola or external genitalia. This finding suggested that *U. simensis* essential oil may not have androgen-related observable adverse effects. In this study, the ossification foci of the sternum, ribs, vertebrae, hyoid, metacarpal, metatarsal, forelimb and hindlimb bones did not show significant differences between gravid rats treated with *U. simensis* essential oil and those in the control groups, both pair-fed and ad libitum. However, the treatment groups had lower number of ossification centers compared to the control groups. This difference was not statistically significant, indicating that *U. simensis* essential oil likely had no observable adverse effects on skeletal ossification. In addition, administration of *U.simensis* essential oil did not affect histopathology of the placenta. This could be the essential oil might not modify the morphology of the placenta or it could be due to regeneration of the placenta's essential oil effect, as the dosage only lasted from day 6 to day 12 of pregnancy. The current study provides scientific evidence about the toxic effects of *U. simensis* essential oil in rat embryos and rat fetuses. However, it was not without limitation. This study did not address postnatal effects of *U. simensis* essential oil. Therefore, it was recommended to investigate the post-natal effects of *U. simensis* essential oil to gain better understanding of its toxicological properties.

## Conclusions

Oral administration of 250 and 500 mg/kg of *U. simensis* essential oil showed no observable adverse effects on the development and growth of rat embryos and rat fetuses. However, rat embryos exposed to 1000 mg/kg of *U. simensis* essential oil exhibited significant reduction in the number of somites and morphological scores compared to the pair-fed and ad libitum control groups. In addition, the in vivo development of the caudal neural tube and forebrain in rat embryos was significantly decreased compared to the control groups. Furthermore, rat fetuses exposed to 1000 mg/kg of *U. simensis* essential oil showed a significant increase in resorption along with a notable decrease in crown-rump length and fetal

weight compared to the pair-fed and ad libitum control groups. Therefore, it was important to inform pregnant women about the potential risks associated with the nutraceutical use of *U. simensis* during pregnancy.

## Supporting information

**S1 Raw data. All raw data support findings of the study.**
(RAR)

## Acknowledgments

We gratefully thank the Ethiopian Public Health Institute for laboratory setup.

## Author contributions

**Conceptualization:** Bickes Wube, Girma Seyoum.

**Data curation:** Bickes Wube, Girma Seyoum.

**Formal analysis:** Bickes Wube, Girma Seyoum.

**Investigation:** Bickes Wube, Kaleab Asres, Samuel Woldekidan, Abiy Abebe, Yonas Girma, Girma Seyoum.

**Methodology:** Bickes Wube, Kaleab Asres, Samuel Woldekidan, Abiy Abebe, Yonas Girma, Girma Seyoum.

**Resources:** Samuel Woldekidan, Abiy Abebe.

**Software:** Bickes Wube, Kaleab Asres.

**Supervision:** Kaleab Asres, Samuel Woldekidan, Abiy Abebe, Yonas Girma, Girma Seyoum.

**Validation:** Kaleab Asres, Yonas Girma, Girma Seyoum.

**Visualization:** Yonas Girma.

**Writing – original draft:** Bickes Wube.

**Writing – review & editing:** Kaleab Asres, Samuel Woldekidan, Abiy Abebe, Yonas Girma, Girma Seyoum.

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
