## [Decision Letter · Decision Letter 0]

PONE-D-25-08532Teratogenic effects of Urtica simensis essential oil in rat embryos and rat fetusesPLOS ONE

Dear Dr. Wube,

Thank you for submitting your manuscript to PLOS ONE. After careful consideration, we feel that it has merit but does not fully meet PLOS ONE’s publication criteria as it currently stands. Therefore, we invite you to submit a revised version of the manuscript that addresses the points raised during the review process.

We look forward to receiving your revised manuscript.

Kind regards,

Eshetie Melese Birru, PhD candidate

Academic Editor

PLOS ONE

2. In the online submission form, you indicated that [The data that support the findings of this study are available from the corresponding author upon reasonable request.].

Additional Editor Comments:

Dear Author,

Thank you for submitting your manuscript entitled “Developmental and Reproductive Toxicity of Urtica simensis Essential Oil in Pregnant Rats.” The study addresses an important public health concern related to the traditional use of herbal remedies during pregnancy and provides valuable preclinical data. Upon editorial review, several strengths were noted, including adherence to OECD guidelines, appropriate animal handling, and comprehensive assessment of embryonic and fetal outcomes. However, to enhance the scientific rigor, translational relevance, and clarity of the manuscript, the following comments and recommendations are provided for your consideration. Overall Significance and Originality

Comments and suggestions:

A. Background and methodology

1. While the study adds valuable insight into potential risks, the novelty is modest unless the authors provide more extensive chemical profiling and mechanistic interpretation.

2. Exposure: The translational relevance to human exposure remains somewhat speculative and would benefit from contextual pharmacokinetic bridging (e.g., comparative doses, route of exposure relevance).

- Why were oral gavage chosen over inhalation or dermal exposure, which may better reflect traditional use (e.g., body fumigation)? The oral route is reasonable for worst-case systemic exposure, but this rationale should be articulated.

3. Use of 5000 mg/kg Limit Dose in Acute Toxicity

- OECD Guideline 425 allows the use of 5000 mg/kg as a limit test only when no prior evidence suggests the substance is toxic.

- However, essential oils often contain bioactive or even cytotoxic compounds (e.g., eugenol, 1,8-cineole, aromatic hydrocarbons), which are mentioned by the authors themselves.

- There’s no chemical safety data presented upfront to reassure that 5000 mg/kg is a biologically plausible or ethical starting point.

- Editorial Suggestion: The authors should:

• Briefly cite prior toxicity reports or justify absence of toxicity warnings from traditional or preclinical data.

• Consider whether lower limit doses (e.g., 2000 mg/kg) would have been more conservative and aligned with OECD recommendations for substances of unknown or expected toxicity.

4. Areas for Improvement:

• The GC-MS chemical profile should be summarized in the current manuscript (even if previously published) to contextualize the toxicological relevance.

• Clarify if the dosing mimics potential human exposure levels (oral vs. inhalation/topical).

• Histological assessment is reported as “no abnormalities,” but quantification or scoring of histological parameters would enhance objectivity.

B. Results and Interpretation

The results are clearly presented with detailed tables and figures. There is a dose-dependent effect, particularly at 1000 mg/kg, suggesting developmental toxicity.

• The absence of malformations with fetal growth restriction and resorption increase implies embryolethality, not necessarily teratogenicity. This distinction should be clarified.

• The discussion occasionally overstates potential mechanisms (e.g., neurotoxicity due to aromatic hydrocarbons) without direct evidence or reference to plasma/tissue levels of these compounds.

C. . Writing and Presentation

The manuscript is detailed but verbose in several sections, particularly the methodology. The writing is generally clear but would benefit from professional editing to improve conciseness and scientific tone.

• Condense repetitive procedural descriptions.

• Improve sentence flow in the discussion to avoid speculative overreach.

• Use standardized toxicological terminology (e.g., “no observed adverse effect level,” “maternal toxicity,” “embryo-foetal development”).

Editorial Decision: Major Revision

Required Revisions

1. Summarize the essential oil composition within the current manuscript.

2. Clarify relevance of dose levels to human exposure and routes of administration.

3. Provide a clearer distinction between embryotoxicity and teratogenicity.

4. Consider including histological scoring or more quantitative assessment of placental findings.

5. Streamline the methodology and discussion for clarity and precision.

6. Moderate speculative mechanistic interpretations unless supported by referenced data.

The study is scientifically sound and offers meaningful insights into the potential risks of Urtica simensis essential oil use during pregnancy. With careful revision to improve clarity, contextual relevance, and scientific balance, the manuscript could make a valuable contribution to ethnopharmacology and reproductive toxicology literature.

Reviewers' comments:

Reviewer's Responses to Questions

**Comments to the Author**

1. Is the manuscript technically sound, and do the data support the conclusions?

Reviewer #1: Yes

2. Has the statistical analysis been performed appropriately and rigorously? 

Reviewer #1: Yes

3. Have the authors made all data underlying the findings in their manuscript fully available?

Reviewer #1: Yes

4. Is the manuscript presented in an intelligible fashion and written in standard English?

Reviewer #1: No

5. Review Comments to the Author

Reviewer #1: Dear authors,

Thank you for being able to read this manuscript. The idea is interesting, but many points need to be changed to improve readers' understanding.

1) The text of the manuscript should be under extensive review of English.

2) The title does not match the manuscript. The authors did not only evaluate the teratogenic effect. They evaluated fetal and placental development, for example.

3) Introduction

- There is no hypothesis. Please insert the hypothesis of this study in introduction.

4) Methodology

- The phytochemical constituents of the plant were not evaluated in this study. However, numerous studies in the literature have investigated the phytochemistry of this species, revealing significant variations in their findings. These discrepancies may arise from differences in several factors, such as the plant parts analyzed (e.g., leaves, stems, or fruits), the geographic origin of the samples, the time of year they were collected, and the extraction methods employed. It is well-established that these variables can significantly influence both the types and concentrations of active principles present.

- Some information must be included, such as geographic coordinates and time of year the plant was collected.

- It is not clear what sample size was used. “five groups of twenty gravid rats”. Twenty rats in each group or 20 in the whole experiment? Then, how many rats were killed on day 12 and how many were left to die at the end of the pregnancy? These informations must be clear.

- Since there was no change in the control groups (group IV – Vehicle and V -ad libitum), couldn't they be placed together and this be reported? This is a suggestion, which could improve the visualization of the results.

- In statistical section, the analysis can be performed only by the software used in this study? If not, remove this information of statistical section. The results of the analysis must be the same regardless of the program used, or even if the calculation are done without any software.

5) Results

- The results could be presented more effectively to enhance data comprehension. A) Control groups (Group I and II) should be presented first in tables and figures. B) Tables should be standardized, with groups consistently placed on the x-axis (or y-axis) across all tables. C) Sample numbers should be included in every table for reference.

- Why was water intake not evaluated in Table 1?

- Some data could be consolidated into single tables for better organization. For example:Tables 5 and 6 could be combined into one table. Similarly, Tables 7 and 8 could be merged into a single table.

6. PLOS authors have the option to publish the peer review history of their article (what does this mean? ). If published, this will include your full peer review and any attached files.

**Do you want your identity to be public for this peer review?** For information about this choice, including consent withdrawal, please see our Privacy Policy .

Reviewer #1: No

---

## [Author Response · Author response to Decision Letter 1]

21 Jun 2025

We uploaded a word file saved as "Authors Response to Editor and Reviewer Comments"

---

## [Decision Letter · Decision Letter 1]

Prenatal developmental toxicity of Urtica simensis essential oil in rat embryos and rat fetuses

PONE-D-25-08532R1

Dear Dr. Wube,

We’re pleased to inform you that your manuscript has been judged scientifically suitable for publication and will be formally accepted for publication once it meets all outstanding technical requirements. However, please ensure that any remaining/irrelevant track changes do not introduce typographical errors or affect the coherence of the text.

Kind regards,

Eshetie Melese Birru, PhD

Academic Editor

PLOS ONE

Additional Editor Comments (optional):

You addressed most of the comments whilst I don't know why you made track changes for no change in some areas (eg., page 13 last paragraph). Please avoid such irrelevant changes and resubmit.

Reviewers' comments:

Reviewer's Responses to Questions

**Comments to the Author**

1. If the authors have adequately addressed your comments raised in a previous round of review and you feel that this manuscript is now acceptable for publication, you may indicate that here to bypass the “Comments to the Author” section, enter your conflict of interest statement in the “Confidential to Editor” section, and submit your "Accept" recommendation.

Reviewer #1: All comments have been addressed

Reviewer #2: All comments have been addressed

2. Is the manuscript technically sound, and do the data support the conclusions?

Reviewer #1: Yes

Reviewer #2: Yes

3. Has the statistical analysis been performed appropriately and rigorously? 

Reviewer #1: Yes

Reviewer #2: Yes

4. Have the authors made all data underlying the findings in their manuscript fully available?

Reviewer #1: Yes

Reviewer #2: Yes

5. Is the manuscript presented in an intelligible fashion and written in standard English?

Reviewer #1: Yes

Reviewer #2: Yes

6. Review Comments to the Author

Reviewer #1: Dear author,

After reviewing the corrected manuscript, I believe the authors have made most of the suggested changes. So, the manuscript can be accept.

Reviewer #2: (No Response)

7. PLOS authors have the option to publish the peer review history of their article (what does this mean? ). If published, this will include your full peer review and any attached files.

**Do you want your identity to be public for this peer review?** For information about this choice, including consent withdrawal, please see our Privacy Policy .

Reviewer #1: **Yes: ** Gustavo Tadeu Volpato

Reviewer #2: No

---

## [Editor Report · Acceptance letter]

PONE-D-25-08532R1

PLOS ONE

Dear Dr. Wube,

I'm pleased to inform you that your manuscript has been deemed suitable for publication in PLOS ONE. Congratulations! Your manuscript is now being handed over to our production team.

Kind regards,

on behalf of

Dr. Eshetie Melese Birru

Academic Editor

PLOS ONE